# Early Emergence of 5′ Terminally Deleted Coxsackievirus-B3 RNA Forms Is Associated with Acute and Persistent Infections in Mouse Target Tissues

**DOI:** 10.3390/vaccines10081203

**Published:** 2022-07-28

**Authors:** Domitille Callon, Anne-Laure Lebreil, Nicole Bouland, Caroline Fichel, Paul Fornès, Laurent Andreoletti, Fatma Berri

**Affiliations:** 1Cardiovir EA-4684, UFR Médecine, Université de Reims Champagne-Ardenne, 51100 Reims, France; dcallon@chu-reims.fr (D.C.); anne-laure.lebreil@univ-reims.fr (A.-L.L.); pfornes@chu-reims.fr (P.F.); landreoletti@chu-reims.fr (L.A.); 2Pathology Department, CHU Reims, Hôpital Robert Debré, 51100 Reims, France; 3Laboratoire d’Anatomie Pathologique, UFR Médecine, Université de Reims Champagne-Ardenne, 51100 Reims, France; nicole.bouland@univ-reims.fr (N.B.); caroline.fichel@univ-reims.fr (C.F.); 4Virology Department, CHU Reims, Hôpital Robert Debré, 51100 Reims, France

**Keywords:** enterovirus, persistent, DBA/2J mouse, 5′ terminal genomic deletion

## Abstract

Major EV-B populations characterized by 5′ terminal deletions (5′TD) have been shown to be associated with the development of myocarditis and type 1 diabetes in mice or humans. To date, the dynamics of EV-B 5′TD-RNA forms’ emergence during the course of infection and their impact on cellular functions remain unclear. Using a RACE-PCR approach in CVB3/28-infected mouse organs, we showed an early (3 days post infection, DPI) emergence of major 5′TD populations associated with minor full-length RNA forms. Viral replication activities with infectious particle production were associated with heart, liver, and pancreas acute inflammatory lesions, whereas clearance of viral RNA without organ lesions was observed in the brain, lung, intestines, and muscles from 3 to 7 DPI. At 28 DPI, low viral RNA levels, +/-RNA ratios < 5 associated with viral protein 1 expression revealed a persistent infection in the heart and pancreas. This persistent infection was characterized by molecular detection of only 5′TD RNA forms that were associated with dystrophin cleavage in the heart and insulin production impairment in beta-pancreatic cells. These results demonstrated that major EV-B 5′TD RNA forms can be early selected during systemic infection and that their maintenance may drive EV-induced acute and persistent infections with target cell dysfunctions.

## 1. Introduction

Group-B enteroviruses (EV-B) are ubiquitous human pathogens transmitted through fecal–oral or respiratory routes. Although most EV-B infections remain asymptomatic, group B coxsackieviruses (CVB) serotypes 1–6 are recognized as common causes of severe acute infection in children and young adults, such as myocarditis, pancreatitis, hepatitis, or meningitis [1]. CVB belong to the genus enterovirus of the family Picornaviridae. Acute EV-B infections can evolve toward persistent infections such as dilated cardiomyopathy (DCM), type 1 diabetes (T1D), and inflammatory myopathy [2,3,4].

EV-B are non-enveloped viruses, with a single-stranded positive-sense RNA genome of approximately 7400 nucleotides [1]. The EV-B RNA genome is flanked at the 5’ end by a highly conserved untranslated region (5′UTR) which is crucial for the initiation of the viral replication and translation activities [1]. In 2008, a coxsackievirus-B2 harboring 5′ terminally deleted (5′TD) populations ranging from 22 to 36 nucleotides was found in the heart of a patient who died of myocarditis [5]. In 2016, Bouin et al. identified by next-generation sequencing (NGS) a majority (99%) of EV-B forms with 5′ end deletions ranging from 15 to 48 nt associated with low proportions (1%) of full-length RNA forms in heart biopsies of a patient with DCM [6]. In 2019, these results were confirmed by NGS in explanted hearts from a cohort of DCM patients [7]. Similarly, in 2005, major 5′TD populations have been found in persistent cardiac infection induced by a cardiovirulent CVB3 strain in a mouse model [8]. These EV-B 5′TD populations were characterized by low ratios of genomic to antigenomic [(+)/(−)] RNA, low production of virions, and low translation activities [6,7,9,10]. EV-B 5′TD populations with 5′ terminal genomic deletions from 8 to 50 nucleotides were found using NGS [7]. Recently, proportions of EV-B 5′TD populations with deletions of 37 to 50 nucleotides have been shown to be negatively correlated with interferon (IFN) expression in peripheral blood of patients with acute myocarditis [11]. By contrast, proportions of the full-length genome or viral RNA forms with deletions of less than 36 nucleotides were positively correlated with IFN expression [11]. In addition, EV-B 5′TD population has also been detected in the murine pancreas at a chronic stage of infection without any detectable cytopathic effect using classical culture assays [3]. Taken together, these findings indicate that EV-B populations characterized by 5′TD ranging up to 50 nucleotides were associated with the development of acute and chronic myocarditis and T1D in mice models or humans [3,7,8]. Moreover, these studies suggested that these EV-B 5′TD forms, similarly to quasi-species, could be early generated and selected during natural infection, and that the emergence of these low-level replicating 5′TD populations may explain how the virus can persist in tissues long after the acute infection leading to chronic pathologies. Indeed, viral proteases have been shown to remain functional even in EV-B 5′TD infected tissues or cells [7,12]. To date, the dynamics of EV-B 5′TD-RNA forms’ emergence during systemic infection and their impact on pathophysiology remain to be investigated.

The 5′ end deletions of EV-B forms have been reported to modify functional secondary structural elements of the RNA domain I (cloverleaf) [8,9,13]. Leveque et al. demonstrated the reduced genomic replicative capacity of these EV-B 5′TD RNA forms which was explained by a disturbed binding of cellular factor PCBP2 (Poly(rC)-binding protein 2) and viral factor 3CD pro (polymerase 3CD) on the 5′UTR domain I secondary structure [9]. EV-B 5′UTR has been shown to be an immunomodulatory element, recognized by RIG-I-like receptors (RLR) [14,15]. Moreover, EV-B 5′UTR has been shown to be a fixation site for the multiple restriction factor of viral replication [16,17,18]. The EV-B with deletions into the 5′UTR could gain a replicative advantage over full-length forms, by the loss of RLR sensing and/or restriction factors fixation sites. Thus, a low IFN response associated with low replicative activities could explain how the virus is detected long after the infection. To date, the impact of EV-B 5′TD RNA forms on inflammatory response profile and more specifically type 1 IFN induction in target tissues remains to be investigated during EV-B acute infection. 

In the present study, we investigated the dynamics of the natural emergence of EV-B 5′TD populations and related inflammatory response profiles (interleukin-10/IL-10, monocyte chemoattractant protein/MCP1, tumor necrosis factor/TNF-α, and IFN-β) during a CVB3/28-induced systemic infection in a previously described DBA/2J model. Using this experimental approach, we analyzed the impact of EV-B 5′TD population emergence and maintenance on inflammatory response profiles, histological lesions, and specialized cell functions in target organs. 

## 2. Materials and Methods

### 2.1. Cells, Virus Strain, and Reagents

Hela cells were grown in minimum essential media (MEM), supplemented by 5 mL penicillin-streptomycin (PS) (Gibco^®^, Paris, France), 5 mL L-Glutamine and 10% Fetal Bovine serum (FBS) (Thermo Fisher^®^, Illkirch-Graffenstaden, France) per 500 mL. The CVB3/28 virus strain was a gift from Steven Stracy (University of Nebraska–Lincoln, Lincoln, NE, USA) [19]. IL-10, MCP1 and TNF-α were purchased from R&D Systems^®^ (Minneapolis, MN, USA).

### 2.2. Virus Production and Titration

Hela cells were seeded at 0.8 × 10^6^ cells per well of tissue culture platelet (six wells) and then incubated at 37 °C overnight. The next day, cell confluence was evaluated at 1.5 × 10^6^ cells per well, based on previous experience; cells were then infected with CVB3/28 at a multiplicity of infection (MOI) of 10^−3^ in free-serum MEM. After 48 h post infection, the supernatant was collected and then clarified using low-speed centrifugation. Virus particles were then measured by plaque-forming unit assay (PFU). Briefly, Hela cells were infected with CVB3/28, making serial dilutions, for 1 h at 37 °C. After viral adsorption, the inoculum was harvested, and cells were overlaid with a medium containing 2% agarose and incubated at 37 °C. After 72 h post infection, viral plaques were visualized using bromophenol blue staining. The viral titers were expressed as plaque-forming unite per 1 mL (PFU/mL).

### 2.3. Ethic Statement

Experiments were performed according to recommendations of the “National Commission of Animal Experiment (CNEA)” and the “National Committee on the Ethic Reflexion of Animal Experiments (CNREEA)”. The protocol was approved by the committee of animal experiments of the Reims Champagne-Ardenne (accreditation number 56) and then by the Ministry of Advanced Education and Research (permits number #5463-201704171500306). All efforts were made to minimize suffering in accordance with the Guide for the Care and Use of Laboratory Animals of the Direction des Services Vétérinaires, the French regulations to which our animal care and protocol adhered.

### 2.4. Infection of Mice

Five-week-old male DBA/2J mice were purchased from Charles River laboratory^®^ (Lyon, France) and housed under pathogen-free conditions. Mice were infected intraperitoneally with 10^6^ PFU/mice in 150µL of physiological serum (saline); uninfected mice were inoculated with 150µL of physiological serum (saline). EV natural infection occurs by oral or nasal routes in a human. For this experimental model, we used an intraperitoneal route to avoid variation in the viral inoculum. To determine survival rates upon infection, loss of body weight was monitored daily for 28 days post infection, and mice were euthanized if they had ≥20% loss of their initial body weight, according to our protocol. Mice were euthanized by cervical dislocation before organ collection at various pre-fixed times post infection (3, 7, and 28 days post infection, DPI). 

### 2.5. Histology and Immunohistochemistry Assays

Infected or uninfected mice were sacrificed at the indicated times. A complete autopsy was performed and organs including the heart, pancreas, liver, brain, lungs, intestines, spleen, and muscles were taken and fixed in formaldehyde 4% for histology. Hearts were sliced into three transverse biventricular sections as recommended for the diagnosis of myocarditis [20]. For each sample, four-µm-thick paraffin sections were cut at three levels. The slides were stained with Hematoxylin, Eosin, and Safran (HES, Pathology department, Reims, France). The slides were examined by a pathologist for cellular infiltration, fibrosis, and necrosis. The lesions were graded using a scale of 0 (no necrosis, inflammation, or fibrosis), 1 (1 or 2 foci by slice), 2 (1 or 2 foci on every slice), and 3 (more than 2 foci on every slice) [21]. In addition, immunostaining was performed for CD3 (1/1000; Dako, Santa Clara, CA, USA, A0452), CD68 (1/100; Dako, M0814), and VP1 (1/1000; Dako, 5D8/1) as previously described [22,23]. 1:200 diluted rabbit anti-insulin (ABclonal, Woburn, MA, USA, A2090), 1:300 diluted rabbit anti-dystrophin (Abcam, Cambridge, UK, ab15277), and 1:500 diluted mouse anti-viral protein 1 (VP1; Dako, 5D8/1) antibodies were used for immunofluorescence staining of fixed heart and pancreas tissue sections. Secondary antibodies were conjugated with AlexaFluor514 and AlexaFluor633 (Thermo Fisher). DAPI (Sigma, St. Louis, MO, USA) was used to stain cell nuclei. Confocal images were acquired using a Zeiss confocal microscope. 

### 2.6. Total RNA Isolation and qRT-PCR

The homogenized organs were subjected to digestion with proteinase K (Merck^®^, St. Louis, MO, USA). Total RNA was isolated with TRIzol RNA Isolation Reagents^®^ (Thermo Fisher^®^) method according to the manufacturer’s instructions. RNA were stored at −80 °C. The viral RNA copy number was quantified by quantitative-RT–PCR with a StepOnePlus Real-Time PCR System (Thermo Fisher Scientific^®^) as described previously, using primers amplifying the CVB3/28: sense (5′-CAC ACT CCG ATC AAC AGT CA-3′), antisense (5′-GAA CGC TTT CTC CTT CAA CC-3′), and probe FAM/TAMRA (5′-CGT GGC ACA CCA GCC ATG TTT-3′) (Eurogentec^®^, Seraing, Belgium) [24]. Viral load was expressed as genome copies per microgram (gc/µg) of nucleic acid extracted. PCR and gel migration were performed as previously described [5,8], using primers amplifying 5′end: S1entB: TTA AAA CAG CCT GTG GGT TGT TCC C; Ext2: GGT TGA TCC CAC CCA CAG G; Ext3: CAT TGG GCG CTA GCA CTC TG. The primer S1entB binds full-length forms. Primers Ext2 and Ext3 bind to 5′ end 15 to 34 nt and to 5′ end 37 to 57 nt (Appendix A). 

### 2.7. Positive- and Negative-Strand RNA Ratio

Negative-strand RNA was isolated from the total RNA molecules by annealing a biotinylated negative-strand specific primer (E3REV; 5′-GGAACCGACTACTTTGGGTGTCCGTG-3′) and binding to streptavidin-labeled magnetic beads (Invitrogen^®^, Life Technologies, Saint-Aubin, France). Purified negative-strand and total viral RNA molecules were quantified with a One-Step real-time RT-PCR assay using serial dilutions of the transcripts for the generation of the standard curves. The positive- to negative-strand viral RNA ratio was then determined using the following calculation: (total EV RNA—negative-strand EV RNA)/negative-strand EV RNA [14].

### 2.8. Rapid Amplification of cDNA Ends-PCR (RACE PCR)

Viral RNA (200 ng) was reverse transcribed using a Superscript II kit (Invitrogen, Thermo Fisher^®^) with 400 nM AvCRev and was incubated for 5 min at 65 °C and then 5 min on ice. The 5′ extremity of cDNA was then ligated with 1× T4 DNA ligase buffer (Ambion, Thermo Fisher^®^), 50 nM Trp1 DNA adaptor, 5 units of Ambion^®^ T4 DNA Ligase (Ambion, Thermo Fisher^®^) was added to the cDNA-mix and was incubated 24 h at 16 °C [6]. Positive cDNA was amplified by a classical PCR reaction using a KAPA Taq polymerase kit consisting of 1× KAPA TaqA buffer, 200 μM dNTP, 400 nM AvCRev, 400 nM Trp1, and 1U KAPA Taq polymerase in a mix of 25 μL with a cycling process composed of one cycle for 3 min at 95 °C, 40 cycles of 30 s at 95 °C, 30 s at 55 °C, 30 s at 72 °C, finishing in a cycle of 30 s at 72 °C. A 2% agarose gel electrophoresis was performed in order to control the length of the PCR products. Amplicons were then sized and quantified by bioanalyzer high sensitivity DNA analyses using the Agilent High Sensitivity DNA Kit (Agilent^®^, Santa Clara, CA, USA) according to the manufacturer’s instructions. Viral populations deleted from less than 8 nucleotides were considered full-length viral populations since the micro-electrophoresis assay has a 10% error occurrence. We used control RNA (synthetic full length or 5′ terminally deleted forms) as a quality check for each run (Appendix A).

### 2.9. Cytokines Quantification

Organs of euthanized mice were harvested, at the indicated times post infection, homogenized in 600 µL of PBS solution with TissueLyser LT (QIAGEN^®^, Hilden, Allemagne), and centrifuged at 12,000 rpm for 15 min at 4 °C. The supernatant was used for: (1) measurement of cytokines by Enzyme-Linked Immunosorbent assay (R&D Systems^®^), (2) virus particles determination as described above. Viral titers were expressed as a plaque-forming unit per mg of total protein (PFU/mg). Measurement of total protein concentration in each supernatant was performed by the Bicinchoninate Acid Method (Sigma Aldrich^®^) according to the manufacturer’s instructions. Cytokine protein level was expressed as pg or ng/mg of total protein by calculating the ratio cytokine concentration/total protein concentration. 

### 2.10. Statistical Analysis

The Mann–Whitney or two-way ANOVA tests were used. Linear regression and Pearson R coefficient were performed. The number (*n*) of animals used in the experiment is specified in the figure legend. All statistical analyses were performed using GraphPad Prism 7 (Prism^®^, San Diego, CA, USA).

## 3. Results

### 3.1. Identification of DBA/2J Mice Target Tissues with Natural CVB3/28 Acute and Persistent Infections

First, we identified target tissues in which an intraperitoneal CVB3/28 infection induced an acute followed by a persistent infection, in DBA/2J mice (Figure 1). DBA/2J mice were chosen for our experiment since CVB3/28 persistence has been reported, and it is a known immunocompetent model for cardiovascular research [22]. During the course of acute CVB3/28 infection (3–7 days post infection, DPI) and at the chronic stage (28 DPI) of viral infection, viral total RNA levels, +/− RNA ratios, viral titers, and VP1 expression were assessed in a large panel of viral target tissues. Total RNA viral loads, VP1 expression, and infectious particle production were maintained at 7 DPI only in the heart, pancreas, and liver, demonstrating an acute infection phase (Figure 1A–D). In other organs, only low levels of viral RNA were observed until 7 DPI, followed by a total clearance of viral markers at 28 DPI (Figure 1A–D). At 28 DPI, low viral RNA levels, +/− RNA ratios < 5 associated with VP1 expression revealed a persistent infection in the heart, pancreas, and liver (Figure 1A–D). Finally, an acute followed by a persistent CVB3/28 infection was observed only in the heart, pancreas, and liver. By contrast, only low levels of viral genomic replication activities were observed in the brain, intestine, muscle, and lung (Figure 1A–D). 

### 3.2. Dynamics of Emergence of 5′TD Viral Populations in Organs of Mice Infected by CVB3/28 Strain

To determine the dynamics of the emergence of 5′TD viral populations in organs of CVB3/28-infected mice, we quantified the number of genome copies of each 5′TD and full-length (FL) viral population at different DPI using a previously validated RACE-PCR approach (Appendix A and Figure 2) [6,11]. Early after the infection (3 DPI), two groups of 5′TD viral forms were characterized: one with deletions ranging from 8 to 36 nucleotides (nt), with the loss of stem a and stem-loop b; and one with deletions ranging from 37 to 50 nt, with the loss of stem-loops a, b, and c of the cloverleaf RNA secondary structures (Figure 2A). Interestingly, these two 5′TD viral populations emerged in all organs during the acute infection phase (3 to 7 DPI) but remained detectable only in the heart, pancreas, liver, and spleen at the chronic stage of infection (28 DPI) (Figure 2B–D). In these organs, we observed a majority of 37 to 50 nt EV-B 5′TD populations with significant viral load means above 10^3^ gc/µg of total nucleic acid extracted until 28 DPI (Figure 2B–D). By contrast, in other organs (brain, intestine, lung, and muscle), we observed an emergence of the same 5′ TD RNA forms without maintenance in these tissues between 3 and 7 DPI (Figure 2C,D). Our results evidenced the early emergence and maintenance of major EV-B 5′TD populations (8 to 36 and 37 to 50 nt 5′ TD RNA forms) during acute and chronic infection phases, without FL RNA forms at 28 DPI only in the heart, pancreas, and liver. 

### 3.3. Inflammatory Cytokines Secretion and Cellular Infiltrates during Acute and Chronic CVB3/28 Infection 

As CVB3/28 5′TD forms emerged and became major, early after systemic infection in all studied organs, we investigated their impact on the inflammatory profile at acute and chronic stages of infection. In the heart and pancreas, CVB3/28 infection induced a strong inflammatory cytokines production, such as IL-10, TNF-α, and MCP-1 (Figure 3B–D). By contrast, the inflammatory response in the liver was weak with no MCP-1 increase (Figure 3C). Several organs without viral genome RNA persistence exhibited different inflammatory response profiles, including IL-10 levels decrease (for the spleen, intestine, and lung tissues) associated with MCP-1 levels increase and no TNF-α levels increase for all of these (Figure 3B–D). Interestingly, IFN-β levels were not increased in any organ (Figure 3A). IFN-α was not increased either (data not shown). Inflammatory cells (T-lymphocytes CD3 and macrophages CD68) infiltrates were found only in the heart, pancreas, and liver tissues (Figure 3E). 

These results demonstrated that the emergence and maintenance of EV-B 5′TD were associated with a specific inflammatory profile, including IL-10, TNF-α, and MCP-1 local secretion and no IFN-β production (Figure 3). In organs without the maintenance of viral genome RNA forms, such an inflammatory profile was not found. Similarly, inflammatory cell recruitment was only found in organs displaying persistent CVB3 5′TD RNA forms. 

### 3.4. Impact of 5′TD RNA Forms on Histological Lesions and Cellular Specialized Functions in CVB3/28 Infected Tissues

To explore the impact of emerging 5′TD RNA forms on organ pathology, histology analyses were performed to detect acute and chronic lesions at 7 and 28 DPI. Massive inflammation and necrosis were found in the heart, pancreas, and liver at 7 DPI (Figure 4A–C). Fibrosis was present only in cardiac tissues at 28 DPI, whereas adipose tissue replacement was found in the pancreas (Figure 4A–C). Inflammation and necrosis were present in the liver at 7 DPI without scarring after 28 DPI (Figure 4A–C). Brain, spleen, intestines, lungs, and muscle were otherwise normal at 7 and 28 DPI (Figure 4B). CVB3/28-induced acute inflammation and necrosis consistent with an acute phase of infection were found in the heart, pancreas, and liver. Lesions consistent with chronic infection were found only in the heart and pancreas (Figure 4A–C). 

Histological lesion scorings were positively correlated with 5′TD viral loads in the heart and the pancreas at 28 DPI (R^2^ = 0.89, *p* = 0.009 and R^2^ = 0.7, *p* = 0.038, respectively; Figure 4D,E). In the heart, at 28 DPI, VP1 was detected in cardiomyocytes by immunofluorescence staining, with colocalization of VP1 expression and dystrophin disruption (Figure 4F). In Langerhans beta cells of the pancreas, VP1 was detected with a decrease in insulin staining or insulin content (Figure 4F). These results indicated that a CVB3/28 persistent infection in target cells—cardiomyocytes and beta cells—associated with chronic histological lesions induced cellular dysfunctions related to pathological processes such as dystrophin cleavage or insulin metabolism impairment. 

## 4. Discussion

In the present study, we investigated how deletions within the 5′UTR affecting the RNA domain I (cloverleaf) structure are related to natural acute and chronic EV-B infections in an immunocompetent mouse model. We evidenced an early emergence of major EV-B 5′TD populations during the acute phase of infection in all studied organs (Figure 2). Major proportions of 5′TD viral populations associated with minor or undetectable full-length viral forms were maintained during chronic infections in the heart, pancreas, and liver of a CVB3/28-infected DBA/2J mouse (Figure 2B,C). Moreover, we showed that these viral RNA forms were related to a chronic persistent infection with histological lesions in the heart and pancreas only (Figure 1, Figure 2, Figure 3 and Figure 4).

The 5′ terminal deletions of RNA viruses (foot-and-mouth disease, Seoul virus, etc.) have been reported, sometimes in association with low virulence or a persistent infection [25,26,27]. RNA-dependent RNA polymerase errors, lack of proofreading, and/or loss of the replication initiation site in 5′UTR are likely to explain the early generation of EV-B 5′TD RNA forms in all study organs (Figure 2B,C) [8,24]. The maintenance of EV-B 5′TD populations was observed only in target organs with histological lesions, whereas proportions of full-length and 5′TD forms were found to be similar and were transiently detected in organs with no lesion (Figure 2B,C). The absence of viral particle detection in the lung, intestines, and brain could be related to specific cell conditions and innate immune response limiting full-length and 5′TD forms replication and translation early during CVB3/28 infection in this model (Figure 1 and Figure 2). These results suggested an evolutionary advantage of 5′TD RNA forms in specific host cell conditions. EV-B 5′TD forms have been shown to be generated only in quiescent (non-dividing) cells in vitro [8]. Differences in cell-type distribution and replication dynamics of EV-B populations could be explained by various expression levels of replication complex cellular factors such as hnRNP C (heterogeneous ribonucleoprotein particle). Their low levels could provide an evolutionary advantage to 5′TD RNA forms over hnRNP C-dependent full-length RNA forms replication, thereby enabling them to become dominant virus populations in target cells: cardiomyocytes, Langerhans beta cells, hepatocytes, or immune cells in the spleen (Figure 2B) [28]. It is well established that the host protein hnRNP C can bind the 5′ termini of full-length poliovirus negative-strand RNA intermediates promoting the synthesis of positive-strand EV RNA [29,30]. Overall, our results suggest that host proteins of RNP complexes of the negative strand 3′ end, possibly hnRNP C, could provide an early replicative advantage to the 5′TD viral forms in specific cells [9,28,31]. Intracellular restriction factors are powerful barriers against viruses [32]. Restriction factors play an important role in the organization of antiviral cellular immunity, with tissue-dependent expression [32,33]. Authors have reported that A3G (APOBEC3G), AUF1 (AU-rich binding factor 1), a cellular protein involved in mRNA stability, and TRIM7 bind to the 5′UTR of various EV-B to limit either viral protein translation or genome replication [16,17,18]. Together with our results, 5′TD viral populations could gain a replicative advantage over full-length forms, by the loosening of restriction factors’ binding sites in specific organ cell conditions. 

During RNA virus infections, the emergence of mutant genomes has been reported and has been defined as viral quasispecies [34]. The 5′TD EV-B genomes fulfilled the criteria of quasispecies, having a lower viral fitness than the complete populations from which it derived [34]. Thus, these 5′TD enteroviral quasispecies became dominant in target cells in our model with sustained viral protein synthesis activities. However, the impact of each 5′TD RNA form on the host cells, or their cooperation or interference with each other, remains to be explored. The sensitivity of the micro-electrophoresis is limited, thus viral populations with a full-length genome could still be present in single target cells, but with such a low level that it is undetectable without high-throughput techniques such as single-cell RNA sequencing or next-generation sequencing [34]. We could not measure the RNA fragmentation burden before the RACE-PCR; however, we used viral synthetic RNA forms as controls (Appendix A).

We identified different inflammatory response profiles in acutely infected organs (Figure 3). The increase in MCP1, TNF-α, and IL-10 levels in the heart and pancreas could be early mediated in a toll-like receptor/TLR3-dependent manner and be responsible for inflammatory cell recruitment and infected-cell-induced apoptosis (Figure 3E and Figure 4A) [35,36]. Sustained IL-10 and MCP1 high levels found in cardiac and pancreatic tissues have been shown to exert an immunosuppressive activity and promote a T helper/Th2 inflammatory response switch resulting in an altered immune modulatory response and potentially promoting a persistent viral infection [37,38]. The inflammatory response in the liver was lower than in the heart or pancreas and was not associated with chronic hepatitis (Figure 3B–D). In the brain, spleen, and muscle, a transient inflammatory response was observed (IL-10 and MCP1), with no histological lesions (Figure 3B–D and Figure 4B). These results suggest that a lower secretion of IL-10 or MCP1 could enhance viral clearance, as previously described by some authors [37,38]. Consistently with these findings, inflammatory cell infiltrates were increased in the heart and pancreas in comparison with the liver (Figure 3E). 

Remarkably, no significant IFN-β level increase was evidenced at the acute phase of infection in all target organs (Figure 3A). The innate immune response mediated by type 1 IFN is a key step in the progression of acute viral infection, which may evolve toward either healing or chronic infection [39]. The importance of this system is evidenced by the variety of mechanisms used by RNA (+) viruses to evade or inhibit IFN induction, signaling, or effector functions and by the various deleterious phenotypes found in humans and mice with genetic disruptions in IFN-β production or signaling [40,41]. Interestingly, 5′UTR deletions have been reported in hepatitis C virus, dengue virus, foot-and-mouth disease, or African swine fever virus, either during persistent infection or with a reduced virulence with a weak IFN response [25,27,42,43]. Recently, 5′TD EV-B with the largest deletions (36–50 nt) have been negatively correlated with IFN response in the blood of acutely EV-infected patients [11]. Consistently, IFN beta therapy has been shown to be effective in EV-induced dilated cardiomyopathy in humans [44]. Together, these findings should stimulate clinical research in immunomodulatory therapies in persistent EV-induced infections and related pathologies. 

Chronic lesions and VP1 expression were found only in the heart and the pancreas, associated with dystrophin and reduced insulin content, respectively (Figure 4D–F). These results indicate that 5′TD viral form persistence with viral protein expression impairs cell functions, such as cytoskeletal disarray and reduced insulin content. Dystrophin degradation has been shown to be the consequence of the viral protease 2A activity, even in 5′TD EV-B infection [45]. Insulin production decrease could be linked to viral protease activity on cellular translator factors [46,47]. Such impairment of specialized cell functions has been shown to be directly linked to the development of dilated cardiomyopathy or type 1 diabetes in mice and humans [48,49,50,51]. 

Such results show that 5′TD viral forms are key drivers in cardiac and pancreatic pathogenesis during EV-B infection. Indeed, dystrophin cleavage is a well-known mechanism of EV-B pathogenicity inducing dilated cardiomyopathy, whereas insulin production decrease is the first step of type 1 diabetes development. Consequently, vaccination against EV-B could have a great impact on public health, years after the vaccination, limiting the development of dilated cardiomyopathy or type 1 diabetes, and efforts should be made to develop such vaccines [52].

## 5. Conclusions

Overall, our results show the natural early emergence of major EV-B 5’TD populations in various organs in mice, and that their maintenance drives the development of a persistent infection of the heart and pancreas associated with specialized cell dysfunctions. Our experimental findings provide important new insights into the role of 5’TD RNA forms as major key pathophysiological factors in EV-B-induced acute and persistent infections of target cells. Further experimental and clinical investigations are needed to confirm our results and to better understand the pathophysiological role of 5′TD populations in EV-B-induced human infections and pathologies. Preventive strategies based on vaccination could limit the development of such persistent infections and chronic pathologies.

## Figures and Tables

**Figure 1 vaccines-10-01203-f001:**
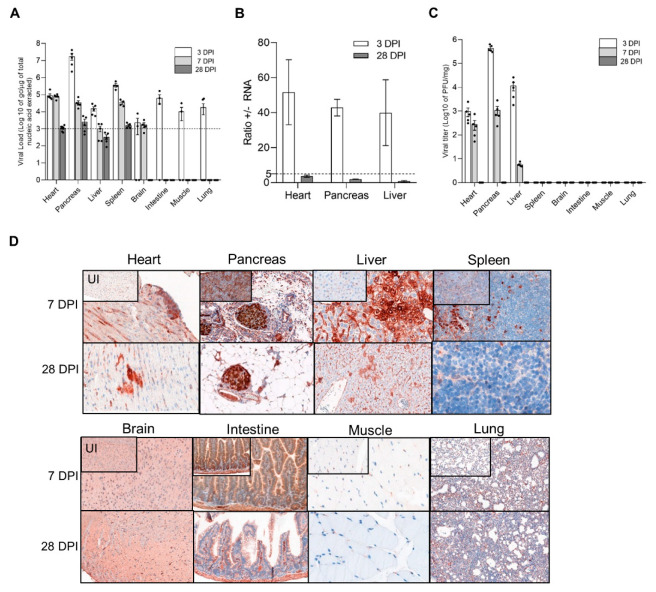
CVB3/28 replication and protein synthesis activities in DBA/2J mouse organs. (**A**) Viral loads were quantified in organs of CVB3/28-infected DBA/2J mice by classic RT-qPCR assay (*n* = 6). Dotted line represents the significance threshold of 10^3^ genome copy (gc)/microgram of nucleic acid extracted (Appendix A). Data represent mean +/− standard error of the mean (SEM). (**B**) Positive to negative RNA ratio (+/− RNA ratio) in the heart, pancreas, and liver at 3 and 28 DPI. Dotted line represents the threshold for viral persistence (*n* = 3). (**C**) Viral titers were measured in organs of CVB3/28-infected DBA/2J mice by classic plaque forming unit assay (*n* = 5). (**D**) VP1 was detected at 7 DPI and 28 DPI in cardiac cells, Langerhans islets cells, and hepatocytes. VP1 was found sparsely positive in the spleen at 7 DPI. Other organs were negative for VP1. Original magnification: 400×. DPI: Days Post Infection. UI: Uninfected.

**Figure 2 vaccines-10-01203-f002:**
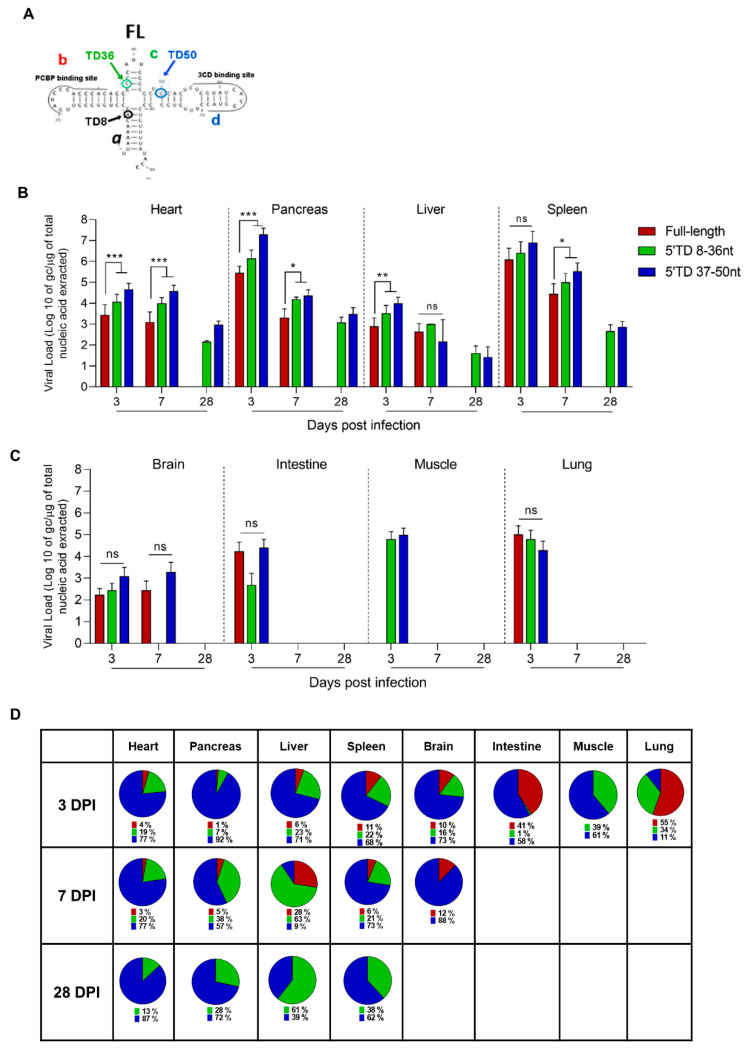
Emergence and maintenance of 5′TD EV-B populations in organs of CVB3/28-infected mice. (**A**) Schematic representation of EV-B cloverleaf secondary structures with stem “a” and stem-loops “b”, “c”, and “d”. Schematic representations of deletions of 8, 36, and 50 nucleotides and its impact on cloverleaf secondary structures, loss of stem “a”, stem “a” plus stem-loop “b”, and loss of stem “a” and stem-loops “b-c”, respectively. (**B**,**C**) EV-B FL and 5′TD respective viral loads were measured using a RACE-PCR method associated with a micro-electrophoresis (Agilent^®^) in organs of CVB3/28-infected mice (*n* = 8 to 10 in B; *n* = 3 to 6 in) (**C**). Data represents mean + SEM (Mann–Whitney U test; *: *p* < 0.05; **: *p* < 0.01; ***: *p* < 0.001, ns: non-significant). (**D**) Schematic representation of full-length and 5′TD populations percentages. Data represents mean. TD: terminally deleted.

**Figure 3 vaccines-10-01203-f003:**
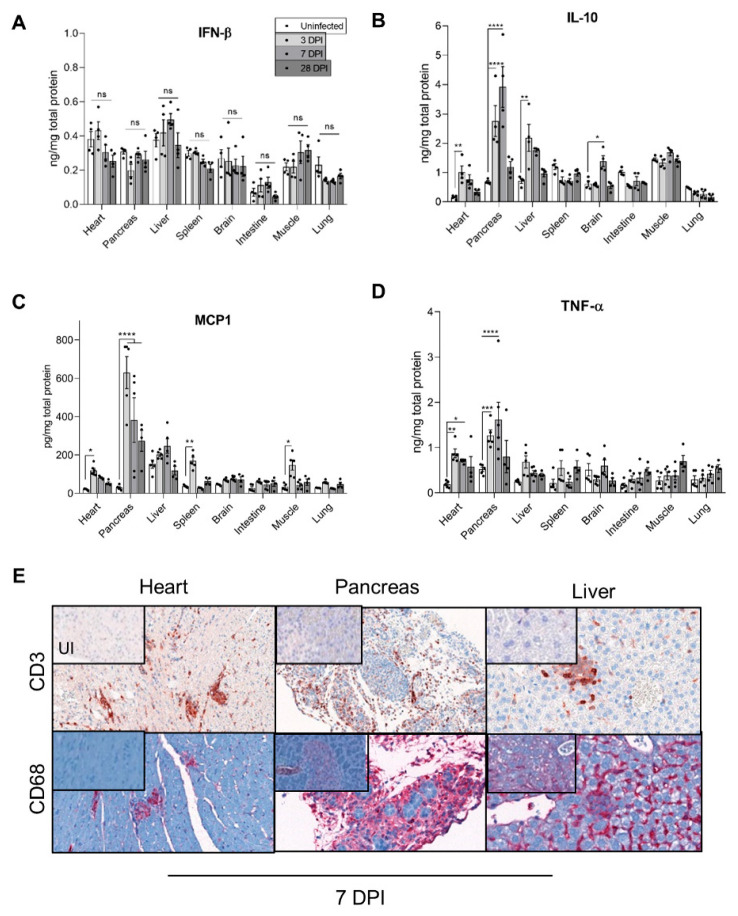
Inflammatory response in organs of CVB3/28-infected DBA/2J mice. (**A**–**D**) IFN-β, IL-10, TNF-α, and MCP1 levels were measured by ELISA in homogenized organs supernatants of CVB3/28-infected DBA/2J mice and compared to cytokines levels in homogenized organs supernatants of uninfected DBA/2J mice (*n* = 4 to 5). Data represent mean +/− SEM (Mann–Whitney U test; *: *p* < 0.05; **: *p* < 0.01; ***: *p* < 0.001; ****: *p* < 0.0001; ns or not specified: non-significant). (**E**) Positive CD3 and CD68 cells were found in inflammatory foci or infiltrates at 7 DPI only in the heart, pancreas, and liver. They were absent in other organs and in uninfected mice. Original magnification: heart, 200×; pancreas and liver, 400×. DPI: Days Post Infection. UI: Uninfected.

**Figure 4 vaccines-10-01203-f004:**
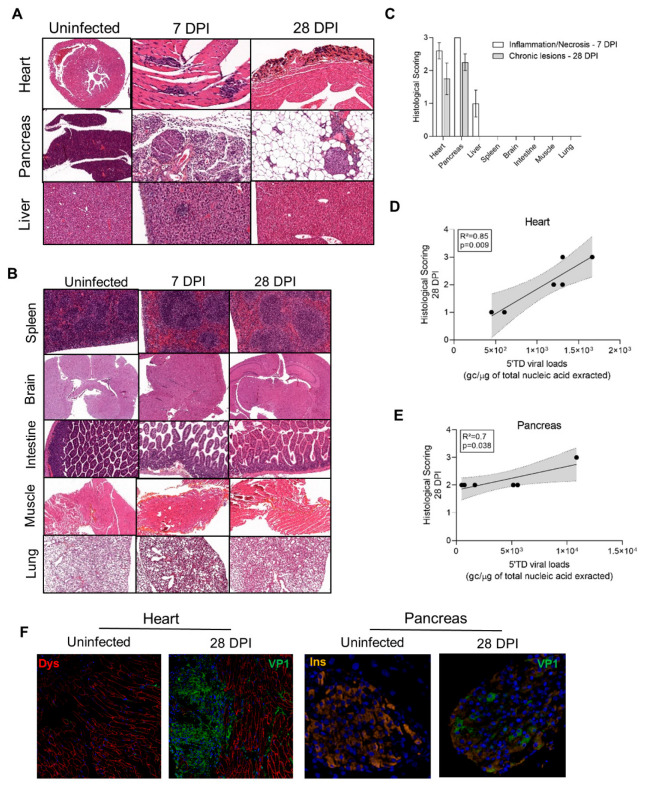
Histological lesions and cell dysfunctions in organs of CVB3/28-infected DBA/2J mice. (**A**) In the infected group, cardiac lesions were found from 7 DPI to 28 DPI. Infected mice exhibited multiple inflammatory foci at 7 DPI. At 14 and 28 DPI, calcified fibrosis was found. At 7 DPI, diffuse pancreatic inflammatory infiltrates with necrosis were found, sparing islets cells. After 14 DPI, an adipose tissue replacement was found, sparing islets cells. Hepatitis was found at 7 DPI, characterized by sparse inflammatory and necrosis foci or sub capsular inflammatory infiltrates. After 14 DPI, no inflammatory or scarring lesions were found. HES, haematoxylin-eosin-safran staining. Original magnification: 10× to 400×. (**B**) Neither acute nor chronic lesions were found in other organs including the brain, spleen, intestines, lungs, or muscle. Original magnification: 10× to 400×. (**C**) Inflammation and necrosis were found in the heart and pancreas at 7 DPI (*n* = 3 to 5). Fibrosis was present only in the heart at 28 DPI (*n* = 4). Inflammation and necrosis were present in the liver at 7 DPI (*n* = 3) only, with no chronic histological lesion. No lesion was found in other organs (*n* = 3). Data represent mean +/− SEM. (**D**,**E**) In the heart (*n* = 6) and pancreas (*n* = 6) of infected DBA/2J mice, a positive correlation between histological lesion scoring at 28 DPI and the 5′TD EV-B load. Linear regression curve, with 95% confidence intervals in grey. (**F**) Confocal analyses revealed a colocalization of VP1 and dystrophin disruption in the heart, and of VP1 and a decrease in insulin staining in the pancreas, at 28 DPI. DPI: Days Post Infection.

## Data Availability

Data are available under request to the corresponding author.

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
