# Peer review of "Early Emergence of 5′ Terminally Deleted Coxsackievirus-B3 RNA Forms Is Associated with Acute and Persistent Infections in Mouse Target Tissues"

_vaccines, 2022, doi:10.3390/vaccines10081203_

Round 1
Reviewer 1 Report
The experimental design is clear, methods are well described, MS is well written. Some abbreviations are unclear (please double-check the definition of all abbreviations). References are up to date. Possibly, one reference to add: Enterovirus RNA sensed through MDA5 followed by PKR-mediated antiviral effects - Gromeier - mbio 2022-06.
It is stated that the DBA/2J strain has been selected since persistent infection by TD enterovirus has been first detected in this strain. Is there any other reason? For instance, these mice are frequently used in cardiovascular research. The DBA/2J strain characteristics include low susceptibility to developing atherosclerotic aortic lesions, high-frequency hearing loss, susceptibility to audiogenic seizures, development of progressive eye abnormalities that closely mimic human hereditary glaucoma, extreme intolerance to alcohol and morphine. Defects of innate or adaptive immunity are not reported.
In general, Figures are too crowded and small. Particularly, histology and fluorescence images in Figures 1, 3 and 4.
Specific points:
63: have been shown to remain functional even...
74: loose; probably LOSS
114: Infection of mice
117, 118: physiological serum; probably saline or PBS
137: antibodies
229: cp/microgram, please define
310: or insulin content (this is seen as immunoreactive insulin in tissue)
329: please specify what is intended by "quiescent cells" (non or slowly replicating?)
382: efficient; change into effective
389: reduced insulin content
Ref. 4 2003 - please check the Introduction (it is mistakenly indicated as 2008)
Ref 7 2005 - as for the previous ones this evidence should be reported as published in 2005
I congratulate the Authors for their efforts in clarifying the biological origin of persisting enteroviruses.
Author Response
Reviewer 1:
The experimental design is clear, methods are well described, MS is well written. Some abbreviations are unclear (please double-check the definition of all abbreviations). References are up to date. Possibly, one reference to add: Enterovirus RNA sensed through MDA5 followed by PKR-mediated antiviral effects - Gromeier - mbio 2022-06.
We thank the reviewer for the positive comments and we agree with the remark. Accordingly, we addressed these concerns in the revised version of the manuscript. In line 396, we added the reference (ref 42) as suggested. We checked the definition of all abbreviations.
It is stated that the DBA/2J strain has been selected since persistent infection by TD enterovirus has been first detected in this strain. Is there any other reason? For instance, these mice are frequently used in cardiovascular research. The DBA/2J strain characteristics include low susceptibility to developing atherosclerotic aortic lesions, high-frequency hearing loss, susceptibility to audiogenic seizures, development of progressive eye abnormalities that closely mimic human hereditary glaucoma, extreme intolerance to alcohol and morphine. Defects of innate or adaptive immunity are not reported.
As mentioned by the reviewer, DBA/2J strain was chosen as it is a frequently used model in cardiovascular research, and it is immunocompetent. We added this explanation in line 204.
In general, Figures are too crowded and small. Particularly, histology and fluorescence images in Figures 1, 3 and 4.
As suggested, we increased the size of histology and fluorescence images when possible in Figures 1, 3 and 4 in the revised manuscript.
Specific points:
63: have been shown to remain functional even... (line 65)
74: loose; probably LOSS (line 77)
114: Infection of mice (line 117)
117, 118: physiological serum; probably saline or PBS (lines 120-121)
137: antibodies (line 139)
229: cp/microgram, please define (line 221)
310: or insulin content (this is seen as immunoreactive insulin in tissue) (line 321)
329: please specify what is intended by "quiescent cells" (non or slowly replicating?) (line 344)
382: efficient; change into effective (line 402)
389: reduced insulin content (lines 406 – 409)
Ref. 4 2003 - please check the Introduction (it is mistakenly indicated as 2008) : line 38, ref was corrected and published in 2008 as indicated.
Ref 7 2005 - as for the previous ones this evidence should be reported as published in 2005 : line 45, ref 8 was indicated as published in 2005
Thank you for these specific remarks, all suggested corrections have been made (cf lines).
I congratulate the Authors for their efforts in clarifying the biological origin of persisting enteroviruses.
We are pleased with these very positive remarks made by this reviewer.
Reviewer 2 Report
The topic in this manuscript would be potentially of interest in molecular evolutional and pathogenesis points of view. However, this reviewer is wondering whether it is relevant to the aims and scope of Vaccines, and the authors did not mention this point in the manuscript.
Author Response
Reviewer 2:
The topic in this manuscript would be potentially of interest in molecular evolutional and pathogenesis points of view. However, this reviewer is wondering whether it is relevant to the aims and scope of Vaccines, and the authors did not mention this point in the manuscript.
This article is a contribution to the special issue of Vaccines: « Molecular Mechanisms of Human Persistent Enterovirus Infections ». Therefore, this report is relevant to the specific aim and scope of the special issue raised by Vaccines. Moreover, the potential impact on vaccination strategies against enterovirus based on our result was added in the discussion and conclusion (lines 418-421 and 430-432).
Reviewer 3 Report
The authors established the group B Coxsackievirus CVB3/28 mouse infection model successfully and detected the 5’ terminal deleted Coxsackievirus-B3 RNA forms at acute or chronic time points after infection. Furthermore, inflammation related cytokines and chemokine were detected in the tissues, T cell and macrophage infiltration in tissues were checked as well. Finally, the lesions of organs from infected mice were visualized by H&E staining. Overall, the novelty of this research is not obvious. Similar data have been published in literature. All the data in this manuscript are clear, but the results such as 5’ TD EV-B forms, cytokines or chemokine production and tissue damage results may not related or support from each other. The authors may raise a specific hypothesis and answer a specific question. Current manuscript is accumulating all the data.
Please describe that Coxsackie B viruses (CVBs) belong to the genus enterovirus of the family Picornaviridae in the Introduction section.
For cytokine or chemokine detection, did the author check IFN-alpha? The authors detected IL-10, this is interesting. Did the author check IL-10 producing B cells?
Fig.3A-D, please describe which two groups to compare for the p values.
Author Response
Reviewer 3
The authors established the group B Coxsackievirus CVB3/28 mouse infection model successfully and detected the 5’ terminal deleted Coxsackievirus-B3 RNA forms at acute or chronic time points after infection. Furthermore, inflammation related cytokines and chemokine were detected in the tissues, T cell and macrophage infiltration in tissues were checked as well. Finally, the lesions of organs from infected mice were visualized by H&E staining. Overall, the novelty of this research is not obvious. Similar data have been published in literature. All the data in this manuscript are clear, but the results such as 5’ TD EV-B forms, cytokines or chemokine production and tissue damage results may not related or support from each other. The authors may raise a specific hypothesis and answer a specific question. Current manuscript is accumulating all the data.
We thank the reviewer for these remarks.
The detection of EV-B 5’TD RNA forms in a few target organs (heart, pancreas) has been reported by us and other teams (A. Bouin et al., « Enterovirus Persistence in Cardiac Cells of Patients With Idiopathic Dilated Cardiomyopathy Is Linked to 5’ Terminal Genomic RNA-Deleted Viral Populations With Viral-Encoded Proteinase Activities », Circulation, vol. 139, náµ’ 20, Art. náµ’ 20, mai 2019 ; K.-S. Kim et al., « 5’-Terminal Deletions Occur in Coxsackievirus B3 during Replication in Murine Hearts and Cardiac Myocyte Cultures and Correlate with Encapsidation of Negative-Strand Viral RNA », Journal of Virology, vol. 79, náµ’ 11, Art. náµ’ 11, juin 2005; S. Tracy, S. et al. « Coxsackievirus can persist in murine pancreas by deletion of 5′ terminal genomic sequences: Coxsackievirus Persistence in the Pancreas », J. Med. Virol., vol. 87, náµ’ 2, Art. náµ’ 2, févr. 2015). However, to our knowledge, dynamics of emergence and maintenance of these forms in other organs after systemic infection of immunocompetent DBA/2J mice have never been reported. The main objective of this study was to investigate for the first time dynamics of CVB3 5’TD RNA forms emergence and distribution in various organs (liver, spleen, brain, intestine, muscle, lung), not limited to the heart and pancreas, after intraperitoneal inoculation of CVB3/28 in DBA/2J mice. Our results showed that CVB3 5’TD RNA forms emerged early after infection in all tested organs, but their maintenance/persistence were observed only in organs with chronic lesions and cells dysfunction (heart and pancreas).
Furthermore, in ongoing work of our laboratory, the direct transfection of synthetic CVB3 5’TD RNA forms in DBA/2J mice showed inflammatory foci in the heart at acute stage of infection. These unpublished results confirm the direct impact and implication of 5‘TD RNA forms in group-B enterovirus-inducedpathogenesis (unpublished data, confidential). These results will be published in another report.
Please describe that Coxsackie B viruses (CVBs) belong to the genus enterovirus of the family Picornaviridae in the Introduction section.
We added this description as suggested (line 32)
For cytokine or chemokine detection, did the author check IFN-alpha?
We checked for IFN alpha production in mouse tissues, and we did not find any increase, as well as for IFN beta. We added this statement as “data not shown” (line 268).
The authors detected IL-10, this is interesting. Did the author check IL-10 producing B cells?
We thank the reviewer for this interesting remark. Indeed, recently it was reported that IL-10-producing B cells attenuate cardiac inflammation by regulating Th1 and Th17 cells in acute viral myocarditis induced by coxsackie virus B3 (Bin Wei, et al, life sci, 2019); however, in our study we did not check specifically for IL-10 producing B cells but it is an interesting suggestion for our future work.
Bin Wei , Yan Deng , Yanlan Huang , Xingcui Gao, Weifeng Wu. IL-10-producing B cells attenuate cardiac inflammation by regulating Th1 and Th17 cells in acute viral myocarditis induced by coxsackie virus B3. Life sci, 2019. doi: 10.1016/j.lfs.2019.116838
Fig.3A-D, please describe which two groups to compare for the p values.
We added a line to precise the compared groups in Fig 3A-D.
Reviewer 4 Report
The authors investigated the course of acute and persistent infections of mice with a coxsackievirus B3 variant. They conclude that deletions at the 5'-end of the virus genome (5'TD) favour virus persistence which is associated with chronic inflammation of heart and pancreas, to a lesser extent of spleen and liver but not of other organs. In a series of papers the authors promote this hypothesis since a few years and many of these papers have been cited here.
I disagree with the authors' conclusion. My main critique is that the authors fail to present convincing evidence of their hypothesis. Detection of 5'-deletions may result from an experimental artefact and may be an overinterpretation of weak data.
General critique:
The authors experimental approach is error-prone. They infect mice with coxsackievirus B3 strain 28 and isolate RNA from various organs at 3, 7 and 28 days post-infection. RNA is reverse-transcribed and cDNA is used to ligate a primer to the 5'-end. cDNA is then amplified. The sequencing procedure is not described here. This experimental approach has pitfalls: (i) The authors cannot ensure RNA integrity during the extraction procedure, RNA molecules may fragment. (ii) Missing 5'-ends of cDNA molecules may result from incomplete RT reaction due to unresolved secondary structures at the very 5'-end of the template RNA. Even if the authors and the Tracy group have published this approach several times, they still fail to refute convincingly the objection of an error-prone experiment. The authors not even discuss the possibility of experimental artefacts or other explanations. Biased referencing and many self-citations apparently support their hypothesis. To my knowledge, equal amounts of plus- and minus-strand RNA in the hearts of persistantly-infected mice was first demonstrated by Klingel et al. (PNAS 89:314-318, 1992) but is not cited here. Instead papers of the Tracy and Andreoletti groups are referenced. In situ hybridization (e.g., sensitive FISH) using suited probes specific to various sites of the 5'-nontranslated region and direct RNA sequencing could be tools to indicate the presence of possible deletions.
Specific comments:
1. lines 28, 38, 89, 238: Do not capitalize enteroviruses, coxsackievirus, penicillin, cloverleaf
2. line 41: explain NGS
3. line 75: explain RLR
4. line 81: explain abbreviations
5. line 88: 5ml penicillin-streptomycin per 100 mL, 1 L or 10 L?
6. line 90: It should read "Steven Tracy".
7. line 94: Is this diminutive appropriate? Usually tissue culture plates are used.
8. line 96: cells per well
9. line 159: Viral RNA was reverse transcribed...
10. lines 185-186: Mann-Whitney's U-test, Pearson R coefficient
11. line 193 and elsewhere in the text: delete "ongoing"
12. Figure 1c: explain the white and grey columns
13. line 211: explain SEM
14. line 220: Did the authors mean "genome copy numbers" or "number of genome copies"?
15. line 229 and Figures 4D, 4E: explain cp
16. line 256: ...were not increased...
17. line 273: functions
18. line 286ff, Legend to Figure 4: Why do authors describe histological lesions at day 3 post-infection if they do not present the results in this figure? They may describe this in the main text if important enough provided with the statement "data not shown".
19. line 293-294: It should read "Neither acute nor chronic lesions..."
20. lines 337-340: Overall, our results strongly suggest... This is a overstatement. No experiment demonstrating restriction of hnRNP C/RNA complexes is presented here. No experiments demonstrate an advantage of 5'TD RNA to full-length RNA are presented. There statement is just a speculation.
21. line 359: explain TLR3
22. line 362: explain Th2
23. line 390-391: Reference 42 is misleading here. Dystrophin degration has been shown by Badorff et al. Nat Med 5:320-326, 1999. Reference 42 is identical to reference 6.
24. Figures presenting histological lesions are too small. Sometimes the resolution is insufficient. Captions to graphs require bigger fonts, especially in Fig. 2D.
Round 2
Reviewer 2 Report
I appreciate the author’s consideration on my previous comments to include several sentences related to the vaccine development in the revised manuscript. For the revised version of manuscript, I have just mention several concerns to be addressed or clarified as follow.- The authors should mention natural oral and experimental ip infections in terms of initial target tissues and dynamics of 5’TD distributions. Is this model susceptible for oral infection?
- At the initial stage of CVB3 infection at 3 days pi, three forms of vRNA (FL, short 5’TD, and long 5’TD) were identified at the same level both in heart and lung for example (Fig. 2B and 2C). But no infectious viruses were detected in the lung (Fig. 1C). The 5’TD emergence and distributions are not simply associated with CVB3 replication and pathogenesis.
・Foot and Mouth Disease and foot-and-mouth disease should be unified. ・Interferon beta should be IFN-β. ・In the Reference section, journal abbreviation should be standardized      (Journal of Virology and J Virol, etc.).
Author Response
Reviewer 2:
I appreciate the author’s consideration on my previous comments to include several sentences related to the vaccine development in the revised manuscript. For the revised version of manuscript, I have just mention several concerns to be addressed or clarified as follow.
We thank the reviewer for his comments.
The authors should mention natural oral and experimental ip infections in terms of initial target tissues and dynamics of 5’TD distributions. Is this model susceptible for oral infection?
As suggested, we added the justification for IP infection over oral natural infection (line 124). We favoured the IP route to avoid variation in viral inoculum. To our knowledge, oral infection with CVB3/28 of DBA/2J mice was not reported.
At the initial stage of CVB3 infection at 3 days pi, three forms of vRNA (FL, short 5’TD, and long 5’TD) were identified at the same level both in heart and lung for example (Fig. 2B and 2C). But no infectious viruses were detected in the lung (Fig. 1C). The 5’TD emergence and distributions are not simply associated with CVB3 replication and pathogenesis.
We agreed with the reviewer that in the lung, 5’TD emergence is not associated with lesions or viral particles detection. We hypothesized that cell conditions and innate immune response in the lung might limit early the replication and translation of CVB3/28 in this model (line 346).
Minor points
・Some viral protein 1 should be VP1. (lines 212, 217, 412)
・Foot and Mouth Disease and foot-and-mouth disease should be unified. (line 404)
・Interferon beta should be IFN-β. (lines 55, 57, 80, 408)
・In the Reference section, journal abbreviation should be standardized (Journal of Virology and J Virol, etc.).
We thank the reviewer for these points; we addressed all of these in the revised manuscript.
Reviewer 3 Report
The manuscript was not improved. The novelty of this research is still not obvious although the authors mentioned that the dynamic of emergence and maintenance of 5' terminal deletion RNA forms have never been reported before.
Author Response
Reviewer 3
The manuscript was not improved. The novelty of this research is still not obvious although the authors mentioned that the dynamic of emergence and maintenance of 5' terminal deletion RNA forms have never been reported before.
We regret that our responses did not convince the reviewer. However, we added substantial supplementary material and we emphasized the novelty of the report for enterovirus pathogenesis, using an experimental model. We could not see which previous publication reported the same results as us.
We agree with the reviewer that the detection of EV-B 5’TD RNA forms the heart and the pancreas has been previously reported (A. Bouin et al., « Enterovirus Persistence in Cardiac Cells of Patients With Idiopathic Dilated Cardiomyopathy Is Linked to 5’ Terminal Genomic RNA-Deleted Viral Populations With Viral-Encoded Proteinase Activities », Circulation, vol. 139, náµ’ 20, Art. náµ’ 20, mai 2019 ; K.-S. Kim et al., « 5’-Terminal Deletions Occur in Coxsackievirus B3 during Replication in Murine Hearts and Cardiac Myocyte Cultures and Correlate with Encapsidation of Negative-Strand Viral RNA », Journal of Virology, vol. 79, náµ’ 11, Art. náµ’ 11, juin 2005; S. Tracy, S. et al. « Coxsackievirus can persist in murine pancreas by deletion of 5′ terminal genomic sequences: Coxsackievirus Persistence in the Pancreas », J. Med. Virol., vol. 87, náµ’ 2, Art. náµ’ 2, févr. 2015). However, to our knowledge, dynamics of emergence and maintenance of these forms in other organs after systemic infection of immunocompetent DBA/2J mice have never been reported. The main objective of our study was to investigate for the dynamics of emergence and distribution of CVB3 5’TD RNA forms in various organs (liver, spleen, brain, intestine, muscle, lung), not limited to the heart and pancreas, after intraperitoneal inoculation in DBA/2J mice.
Our descriptive results clearly showed that CVB3 5’TD RNA forms emerged early after infection in all tested organs, and that remarkably their maintenance/persistence without full-length viral RNA forms were observed only in organs with chronic lesions, associated with cells dysfunctions (heart and pancreas).
In summary, the EV-B persistent infection was characterized by the detection of only 5’TD RNA forms associated with dystrophin degradation in heart and insulin content decrease in beta pancreatic cells. Our results showed that major EV-B 5’TD RNA forms can be early selected during a systemic infection and that their maintenance may drive EV-induced acute and persistent infections with target cells dysfunctions. This is the main scientific novelty presented in the manuscript.
Round 3
Reviewer 3 Report
No more comments. Thanks.